# Re-Os Age and Stable Isotope (O-H-S-Cu) Geochemistry of North Eastern Turkey's Kuroko-Type Volcanogenic Massive Sulfide Deposits: An Example from Cerattepe-Artvin

Ali Ucurum [1], Cigdem Sahin Demir [2], Nazmi Otlu [1], Mustafa Erturk [1], Taner Ekici [1], Jason Kirk [3], Joaquin Ruiz [3], Ryan Mathur [4,*] and Greg B. Arehart [5]

1 Department of Geology, Sivas Cumhuriyet University, 58140 Sivas, Turkey; aliucurum@cumhuriyet.edu.tr (A.U.); notlu@cumhuriyet.edu.tr (N.O.); mustafaerturk05@hotmail.com (M.E.); tanere@cumhuriyet.edu.tr (T.E.)
2 Technical Vocational School, Sivas Cumhuriyet University, 58140 Sivas, Turkey; csahin@cumhuriyet.edu.tr
3 Department of Geosciences, University of Arizona, Tucson, AZ 85712, USA; jdkirk@arizona.edu (J.K.); jruiz@email.arizona.edu (J.R.)
4 Department of Geology, Juniata College, Huntingdon, PA 16652, USA
5 Mackay School of Earth Science and Engineering, University of Nevada, Reno, NV 89557, USA; arehart@unr.edu
* Correspondence: mathurr@juniata.edu; Tel.: +1-814-641-372

**Abstract:** The eastern Pontide tectonic belt (EPTB) contains greater than 350 identified Kuroko type volcanogenic massive sulfide deposits/mineralization/occurrences (VMSD). The deposits are associated with Late Cretaceous felsic volcanics consisting mainly of dacitic and rhyolitic lavas and pyroclastics that outcrop within a narrow zone running parallel to the eastern Black Sea coast and represent the axial zone of a paleo-magmatic arc. The Cerattepe deposit is the second-largest and is a hybrid VMS system with some epithermal features. To date, no geochemical research constrains the origin and timing of mineralization in the Cerattepe VMS deposit. Here, we provide Cu, O, H and S, isotope analysis of ores and alteration minerals to understand the hydrothermal history of the deposit and date the massive ore with Re-Os geochronology. Secondary weathering mobilized and redistributed metals in the deposit. The copper isotope signatures of shallow ores in the gossan follow patterns resulting from oxidative weathering of copper minerals with gossan Fe oxides of $\delta^{65}Cu = -2.59‰$, enrichment zone copper sulfide of $d^{65}Cu = +2.23$ and $+1.73‰$, and primary ores of $\delta^{65}Cu = +0.71$ and $+0.41‰$. At the boundary of the enrichment zone, further cycling and migration of enrichment zone copper are evidenced by two samples having larger ranges of the $\delta^{65}Cu = +3.59‰$, and $-2.93‰$. Evidence for a magmatic source for fluids and S are evidenced by the O and H isotope values from quartz veins ($\delta^{18}O = +7.93‰$ to $+10.82‰$, and $\delta D = -78‰$ and $-68‰$) and sulfides that possess $\delta^{34}S$ ratios of $-5$ and $0‰$ from drill core samples. $^{187}Os/^{188}Os–^{187}Re/^{188}Os$ ratios from drill core sulfide samples of Cerattepe VMS deposit yields a $62\pm3$ Ma isochron age and a highly radiogenic Os initial ratio. This age is compatible with silicate alteration ages from a proximal deposit and clearly shows mineralization occurs at a much younger time than previously proposed for VMS mineralization in the eastern Pontides. The new Re-Os age and source of Os imply that mineralization in the area occurs at a distinctly younger interval in the back-arc basin and metals could be sourced from the surrounding host rocks.

**Keywords:** Cerattepe-Artvin VMSD; O; H; S; Cu isotopes; Re-Os geochronology; Turkey

## 1. Introduction

Part of the Alpine-Himalayan belt, the eastern Pontides tectonic belt (EPTB) is located in the northeastern section of Anatolia. The subduction, accretion, and collision events associated with the closure of the Tethyan Ocean caused the geological structure

of the EPTB [1–4]. Because of a lack of geological, geochemical, geochronological, and paleontological data, the geologic history of the area is hotly debated.

Several models have been proposed to explain this geologic history. First, the eastern Pontides orogenic belt formed as a result of the northward subduction of Paleotethys during the Paleozoic-Mesozoic period [5–9]. Second, the idea that the eastern Pontides orogenic belt was created by the southward subduction of the Tethys oceanic lithosphere located in the northern section of the orogenic belt during the Paleozoic-Mesozoic-Cenozoic period and that the Black Sea is a remaining fragment of the Tethys ocean [10–14]. The third model [15] suggested that the Paleotethys was located north of the Pontides, and southward subduction operated from the Paleozoic until the early Mesozoic periods, followed by northward subduction from the late Mesozoic until the early Cenozoic [16].

The volcanogenic massive sulfide (VMS) deposits of Turkey are found in three tectonically different regions: the Kuroko-type deposits in the eastern Pontides tectonic belt (EPTB), the Cyprus-type VMS deposits in both the Kure area (Kastamonu) and along the Bitlis-Zagros Suture Zone (BZSZ), and the Besshi-type VMS deposits in the Hanönü–Tas-Taskopru area (Kastamonu) (Figure 1). Deposits occurring on the northern side of the Pontides (Murgul, Çayeli, Kutlular, Harköy, and Lahanos) have been thoroughly studied, but the deposits on the eastern side have been mostly ignored (Figure 1).

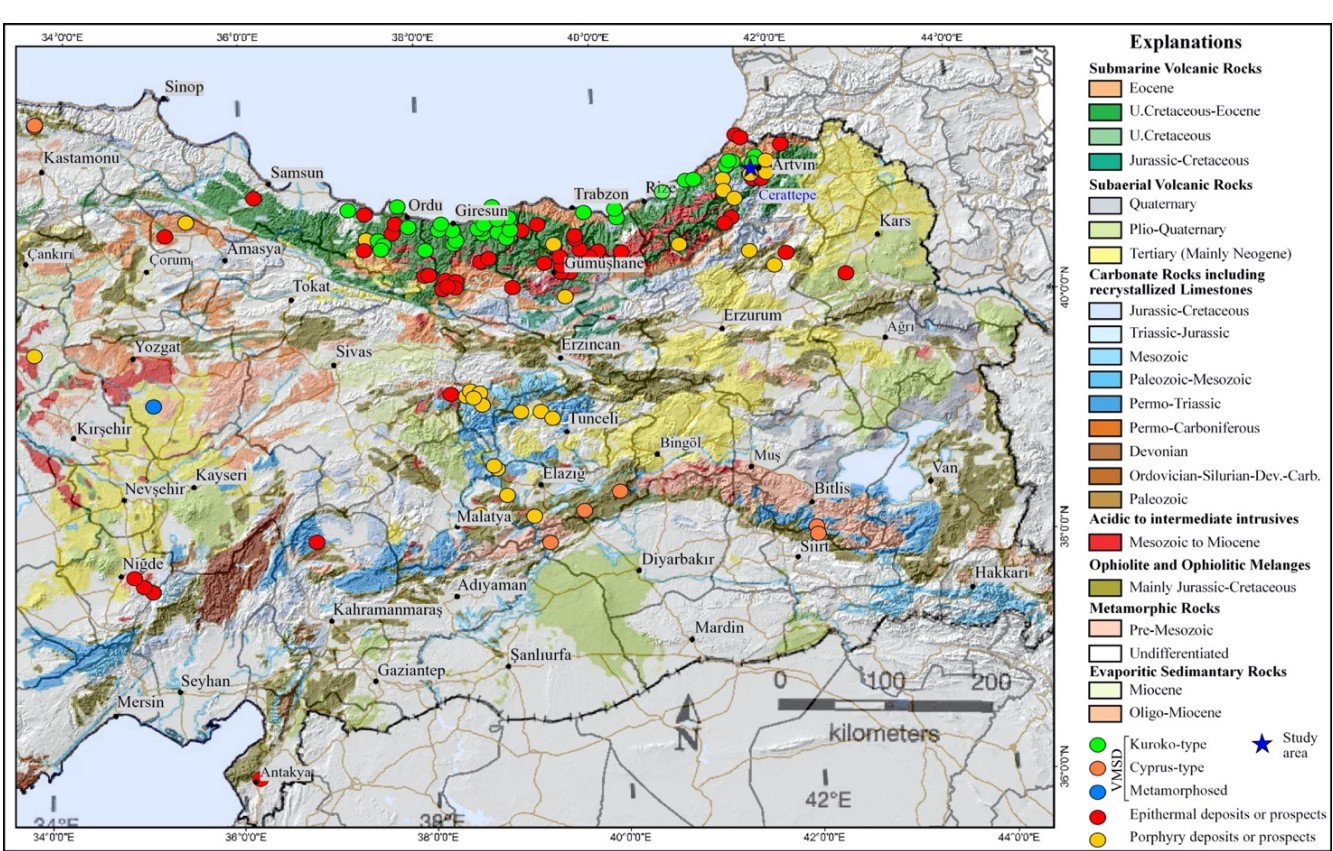

**Figure 1.** Geological map with continuous SRTM topography surface and distribution of the VMS, Epithermal, and porphyry deposits and prospects of Central-Eastern Turkey with emphasis on host rock lithology. Geology and location of deposits are from [17].

All of the Kuroko type VMS deposits are associated with Late Cretaceous felsic volcanic rocks primarily consisting of dacitic and rhyolitic lavas and pyroclastics that are found in outcrops as a narrow zone running next to the eastern Black Sea coast; these represent the axial zone of a paleo-magmatic arc. Cu-, Zn-Cu-, Cu-Zn-, and Zn-Pb-Cu metal associations are subtypes of these deposits.

A number of the Kuroko-type VMS deposits are located in the Eastern Pontides Tectonic Belt (EPTB) in northeastern Turkey [12,13,16–21]. One of the four major tectonic units that form Anatolia is the EPTB, a complex structure of island arc terrains that evolved during the subduction of the Afro-Arabian plate beneath the Euro-Asian plate and the resulting closing of the northern branch of the Tethys Ocean [3,4]. The EPTB is a part of a larger metal-rich tectonic band that stretches from southern Georgia and northern Armenia to Bulgaria and Romania.

There has been only limited research on these deposits despite the long history and exploration of the VMS deposits in Turkey. While no significant discoveries have been made since the Cerattepe Cu-Au VMS deposit in the early 1990s, there has also been renewed interest in the old, abandoned mines of the eastern Pontides. These are under development and pre-production at this time. The Cerattepe (Figure 2) discovery is a very important discovery in terms of the VMSD in the last 50 years in Turkey's exploration program which has been conducted by different foreign exploration-mining companies and their local joint-ventures.

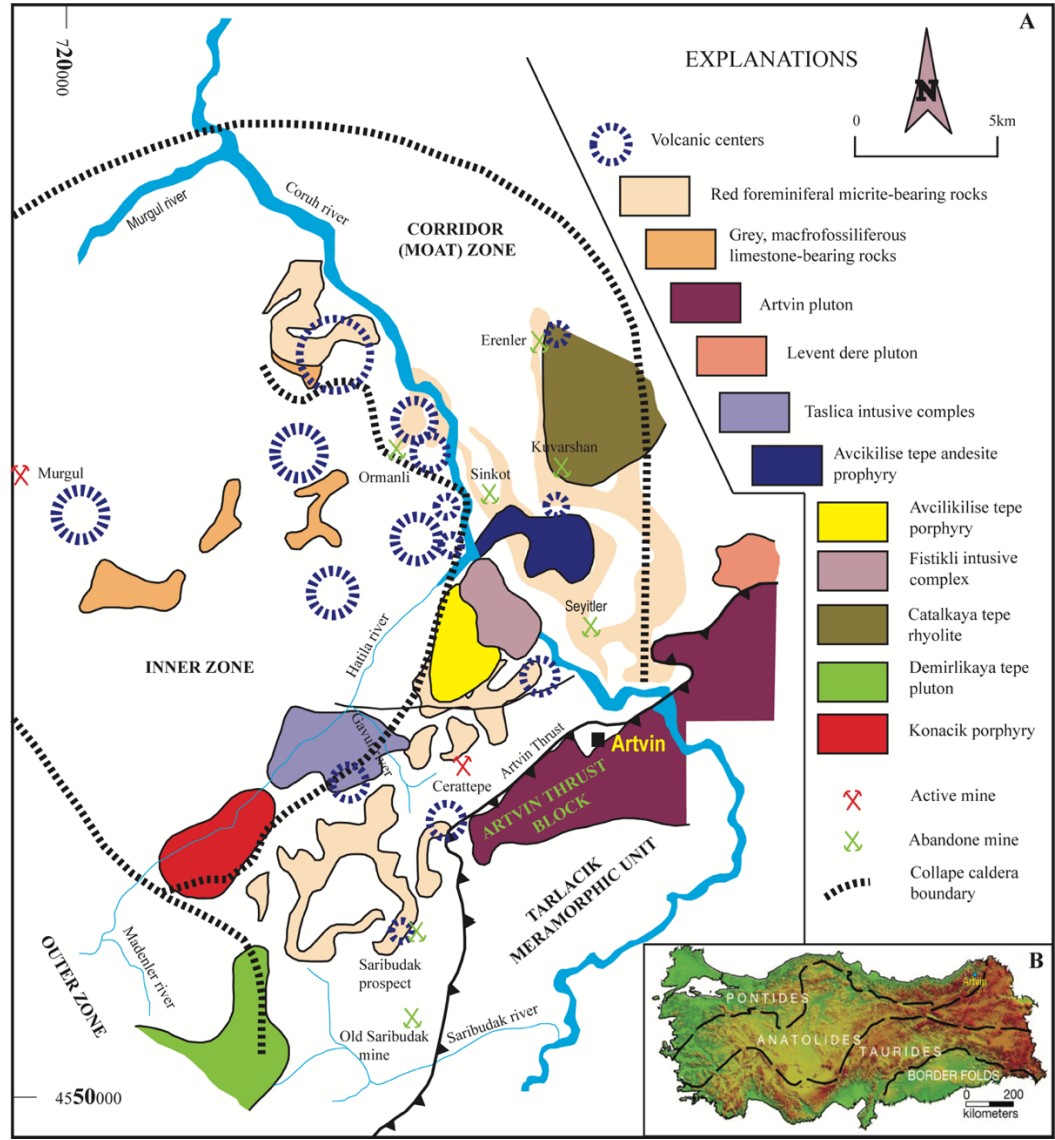

**Figure 2.** (**A**) Simple geological and location map of VMSD in Artvin area and volcanic architecture and surround intrusion and volcanic rocks with volcanic centers, modified from [22], (**B**) Main tectonic units and location of Artvin province after [1].

The Cerattepe deposits (Figure 2) are not well studied except for [22] and no published papers focus on these deposits. These studies have reported the nature and timing of the mineralization in the area since its discovery.

To determine the specific ore genesis of one deposit, we present an integrated set of isotope analyses that characterize the nature of the hydrothermal system, timing of mineralization, and supergene weathering profile. O and H isotope analysis (O, H) of veined quartz and S isotope of massive sulfide ores characterize the high-temperature ore-forming solutions. Re-Os geochronology from sulfide minerals (pyrite, chalcopyrite) from massive ore record the timing of high-temperature mineralization and source of Os and by inference the metals of the deposit. The $\delta^{65}$Cu isotope ratios in sulfide ore minerals (chalcopyrite, pyrite, etc.) from gossan and massive ore zone show the cycling of the copper due to uplift and subsequent oxidative weathering of the massive ores.

## 2. Materials and Methods

Sampling was done between 2002–2005 with NATO and DPT (Turkish State Planning Organization) grants. To conduct O, H, and S isotope analysis and Re-Os geochronology 11 samples from silicified gossan zone and 10 ore samples from drill cores CTD-111, CTD-117, and CTD-190 of Cominco-Turkey were collected. Samples were crushed and minerals were handpicked for analysis.

Analysis of stable isotope rates ($\delta^{18}$O, $\delta$D, $\delta^{34}$S) was completed at Nevada Stable Isotope Laboratory, University of Nevada, Reno (Reno-NV, USA), and $\delta^{65}$Cu isotope were measured from these samples on an MC-ICP-MS at Pennsylvania State University (State College, PA, USA)

$\delta^{18}$O value of quartz has been analyzed with a continuous flow method by using the laser-analysis technique where $BrF_5$ is used for fluoridation. The method being used has been obtained by making changes with the methods [23]. Samples have been warmed with Merchantek EO $CO_2$ laser and isotope analysis has been made with molecular $O_2$ at Micromass Isoprime isotope ratio mass spectrometer with double entries. Analysis has been calibrated by using the standard having no NBS-28 ($\delta^{18}$O = +9.6‰). The error ratio is ±0.15‰. Values have been reported as per V-SMOW (Vienna Standard Mean Oceanic Water).

$\delta$D the liquid inclusions within quartz analyzed by using the Micromass Isoprime isotope ratio mass spectrometer with a continuous flow line [23]. Analysis has been calibrated by using Acetanilid ($C_8H_9NO$) standard ($\delta$D = −130‰). Error ratio is ±2‰. Values have been reported as per V-SMOW (Vienna Standard Mean Oceanic Water).

For determining $\delta^{34}$S isotope value in 4 whole-rock samples, firstly S has been enriched by using the Kiba method [24] and the $\delta^{34}$S ratio has been determined with Micromass Isoprime isotope ratio mass spectrometer by using the method [25]. Analyses calibrated by using GSL (Green Sphalerite), UGLI (Galena), BSL (Brown Sphalerite), and MIC (Chalcopyrite) standards. The error ratio is ±0.2‰. Values have been reported as per V-CDT (Vienna Canyon Diablo Troilite).

$\delta^{34}$S analysis on 6 samples containing sulfur minerals has been analyzed utilizing direct combustion at 1000 °C at Eurovector 3000 model element analysis device. S gas is transformed into $SO_2$ gas in the furnaces where oxygen fugacity is buffered with Cu-CuO mixture. Analysis has been made at the established $SO_2$ gas chromatography with Micromass Isoprime isotope ratio mass spectrometer with purified and continuous flow line. S isotope analysis has been carried out by using the method similar to that of [25]. Analysis has been calibrated by using GSL (Green Sphalerite), UGLI (Galena), BSL (Brown Sphalerite), and MIC (Chalcopyrite) standards. Error ratio is ±2‰. Values have been reported as per V-CDT'e (Vienna Canyon Diablo Troilite).

Samples for Cu isotope were separated with hand picking and crushing. The processing of the sulfide samples was conducted following [26] where the sulfide powders were dissolved in Teflon beakers with 4 mL of ultra-pure aquaregia at 140 °C for 24 h. These solutions were dried and then the copper concentrate of copper was measured from them

on an ICP-OES at Pennsylvania State University to determine the amount of the solution to dry for purification by ion-exchange chromatography. Approximately 10 micrograms of copper were used for ion-exchange chromatography defined in [27].

Copper isotope ratios were measured on the Neptune (Thermo brand) at Pennsylvania State University in State College. The instrument was in low-resolution mode and the solution concentrations were at 100 ppb. This generated a 5 V signal on $^{63}$Cu. A correction for mass bias was completed by the use of standard-sample-standard bracketing with the NIST 976 international isotope standard. One block of 30 ratios was measured for each sample and this procedure was repeated so that the reported values are an average of 60 ratios. The reported values are in the traditional per mil format compared to the NIST 976. Errors for the analyses are 0.08‰ (2σ) and are calculated by the overall variations of the NIST standard for the duration of the measurement session [27]. Replicate values all fall with this reported range. An internal USA cent from 1838 had a $^{65}\delta$Cu = −0.02‰ within the error of the value reported in [26].

Re and Os concentration and isotope ratio of sulfide ore samples were measured with a Negative Thermal Ionization Mass Spectrometer (NTIMS) at University of Arizona Tucson. Samples with quantities varying between 0.6 and 1.5 g have been heated for 18 h at temperature levels above >210 °C in carius tubes by using ultra-pure reverse aqua-regia process by applying [28] method. As described by [29], by adding spike and acid together with 2–3 mL hydrogen peroxide, oxidation and sample spike equilibration were achieved. To further enhance oxidation of Os, 3 mL of hydrogen peroxide was added before the distillation process [30]. Following this, Os was purified with a micro distillation device and Re was purified using two stages of ion change chromatography [30]. The largest source of error in the analysis is the blank contribution of Os. The blank has been measured as $^{187}$Os/$^{188}$Os = 0.24 for 0.9 and 34 picograms of Os and Re respectively [30]. Age has been measured by using Isoplot version 4.15 (K.Ludwig) for Windows by using $^{187}$Os/$^{188}$Re isochron method against $^{187}$Re/$^{188}$Os. 2 sigma error ratio has been determined during age calculation.

## 3. Cerattepe VMS Deposit

Upper Cretaceous volcanic-volcanoclastic and igneous rocks hosted ore deposits are common in the Eastern Black Sea region (Figure 1).

### 3.1. Introduction

Cerattepe VMS deposit is composed of two different deposits. These are the main deposit and Kardelen deposit. Both of the deposits have different hydrothermal exit centers (Figure 1b). While the main deposit is located at upper elevations, the Kardelen deposit is located 100–150 m below current altitude and at a distance of 500 m from the main deposit. Mostra known as Cerattepe lies 100 m to the north of the main deposit. While the main deposit is economical with high grade and reserve at (3.9 mt, % 5.2 Cu, % 2.0 Zn, % 0.5 Pb, 1.2–4 g/t Au, 25–140 g/t Ag), Kardelen deposit is not economical and no samples were analyzed from this deposit (0.4 mt, % 5 Cu, 2 g/t Au ve 90 g/t Ag) [22].

### 3.2. Geology

First preliminary geological study is conducted by MTA geologist [31], and later on detailed geological study was completed by a Ph.D. study at the Cerattepe area [22] has revealed the presence of Kuroko type volcanogenic massive deposit which developed along NE direction in massive and in disseminated form, within Upper Cretaceous Dacitic tuffs host rock concerning the detailed geology and mineralization of the area.

The Cerattepe deposit is Cretaceous in age [16,31], with volcanic-sediment and intrusive host rocks, having the length of 440 m and being composed of lensoidal pyritic breccia that can reach 84 m thickness locally and increase in up to 180 m [32,33]. As a result of the reserve study being realized in 1996, 4.1 m/t reserve has been determined and % 4.9 Cu, 1 g/t Au, 21 g/t Ag grade has been determined. Cerattepe deposit has been distinguished

as a high-grade zone at the base and a low-grade zone that followed it. It has been found out that at Au-Ag gossan zone which overlapped Volcanogenic massive sulfide deposit, there was a probable reserve of 7.3 million tons and that it contained 4.1 g/t Au and 145 g/t Ag grade.

The Cerattepe VMS is a hybrid deposit formed during rapid volcano-tectonic rise during under-sea–under-surface transition [34,35]. The main deposit is the base zone and it has high-grade copper sulfur and oxide zone being rich with Au+Ag+Pb+Ba present in the overlapping cover. The Cerattepe VMS deposit has characteristic features being unique to typical Kuroko [36,37] or bimodal felsic [38–40] VMS deposits. An oxide zone rich zone with gold is characterized by the deposition of primary oxide minerals and hydrothermal oxidation that developed from the oxidation of primary sulfur minerals or with the current alteration of sulfur minerals.

### 3.2.1. Massive Ore

The Cerattepe deposit has similar features with the Cayeli-Rize deposit which is operated by Cayeli-Bakir isletmeleri A.S. However, the unique difference of Cerattepe copper deposit is that gossan zone developed on the top of VMS deposit possesses high gold and silver content (like Hod deposit in Artvin). It is predicted that these processes enable chalcocite and other minerals to precipitate at a specific zone at the base of massive sulfides. As a conclusion, this mineral deposit is recognized with high-grade copper sulfides and the oxide zone is rich with gold, silver, lead, and barium which overlaps and surrounds it.

Cerattepe Formation (Hatilla River member and Seyitler member) constitutes the base rock of mineral deposit [31]. On the mineral zone, there are gray lapilli tuffs, red siltstone, mudstone and sandstone, and pelagic limestone lenses of Coruh Formation. Hanging wall rock series has intruded by the feldspar porphyry basaltic dykes and sills which feeding the basaltic lava on top (Figures 3–5).

The mine deposit is within the syn-volcanic depression basin (graben) limited to the northeast-oriented faults. This structure is approximately 1000 m long and 150 m wide. The rhyolitic dome and rocks that have been subjected to severe silica and pyrite alteration events constitute the southeastern side of the basin.

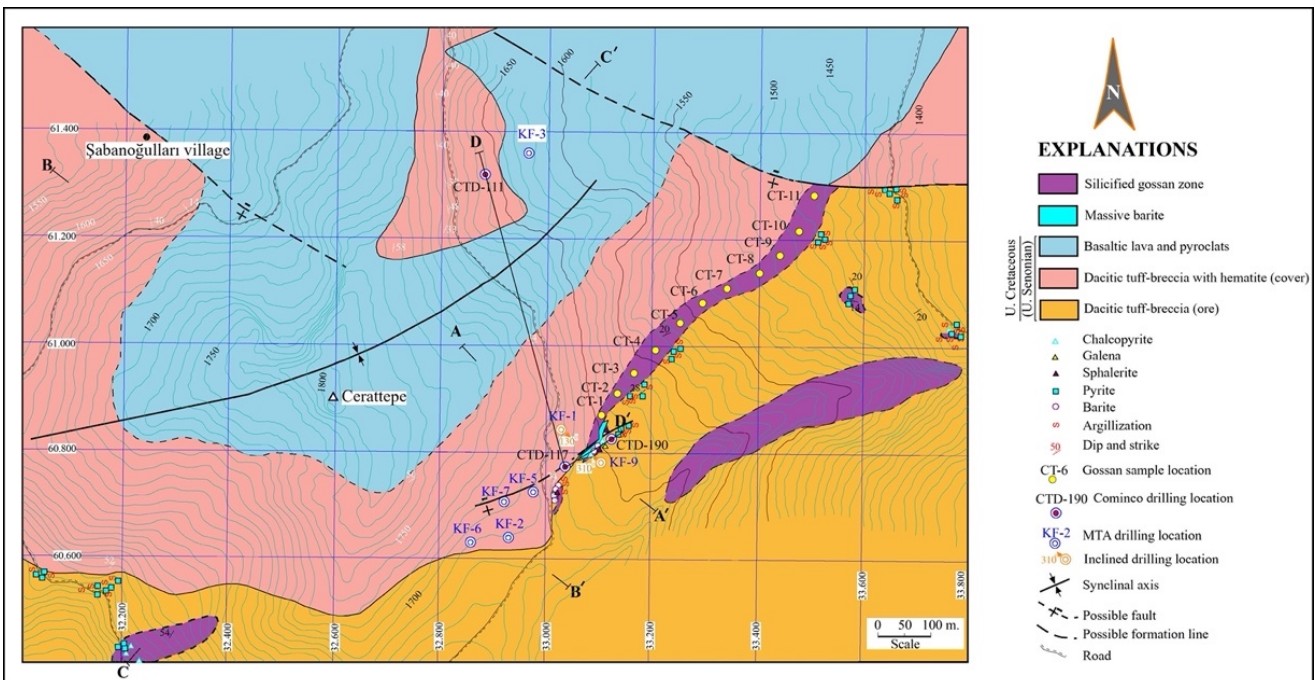

**Figure 3.** Geological map of the Cerattepe-Artvin area, modified after [31].

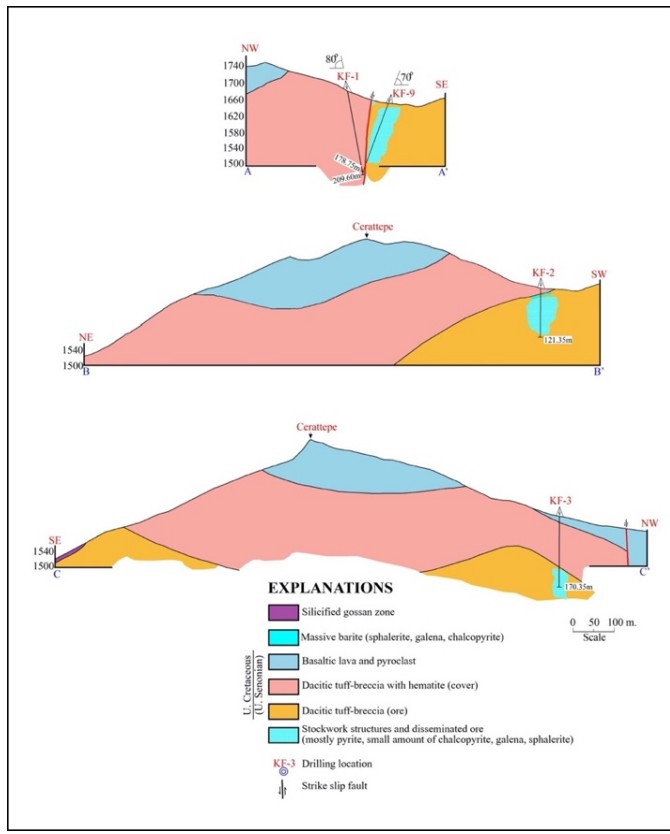

**Figure 4.** Cross-sections the Cerattepe-Artvin area, modified after [31].

Syn-volcanic faults in North-South and North-East directions, constitute the west and north borders. The sulfide zone of mine deposit has a thickness of 70 m in the form of a lens and it has a width of 100 m. This zone has a 320 m direction and contains solid-brecciated, base metal-rich, sulfide minerals, and pyrite. These sulfides have been subjected to deep degradation, leaching, and secondary enrichment factors. Zoning from low copper grade towards high copper grade being observed in the mineral deposit can be connected with primary zoning and secondary enrichment that follows.

Sulfides occur in two different zones. High-grade zones of mineral deposits in the form of the lens (>5% Cu) constitute the deposit base. Material with lower grade is present at the top of this zone and partially as surrounding the base zone. The border between these two different zones is a transition zone of a few meters being irregular. The biggest one among high-grade lenses has a thickness of 5–20 m and width up to 90 m and a length of 300 m (Figures 4 and 5).

The main ore mass of the deposit contains high-grade copper sulfides having a massive structure and there is an oxide zone rich with gold, silver, and barium, being characterized with dense alteration stages on top of these [41]. At the deposit where substitution took place, cataclastic, annealing, separation, colloform, framboidal, and bird's eye textures have been observed. Pyrite, sphalerite, marcasite, chalcopyrite, bornite, galena, tennantite-tetrahedrite, gold, covellite, digenite, chalcopyrite cuprite, and cubanite have been observed as ore minerals and quartz, calcite and barite have been observed as gangue [41]. At the oxidation zone, limonite, hematite, malachite, azurite, and jarosite have developed (Figures 6–10).

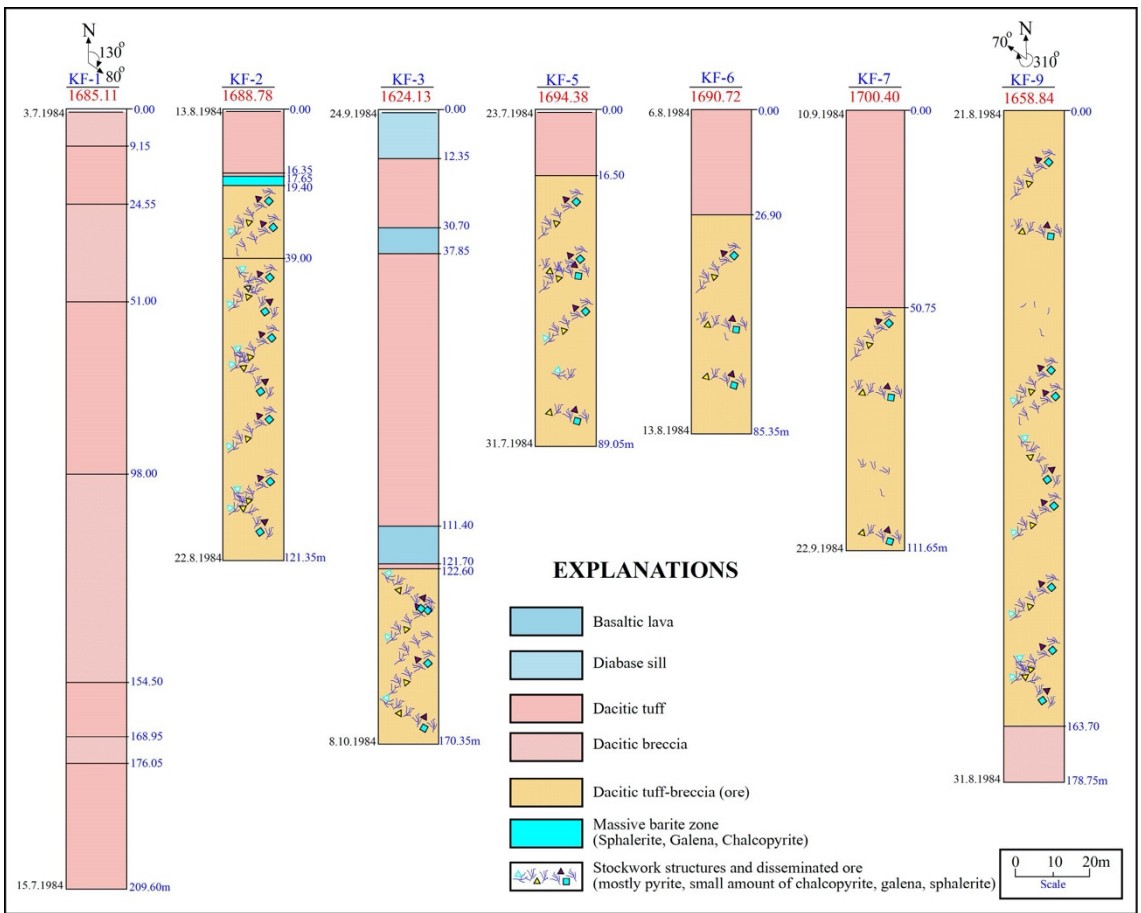

**Figure 5.** Drill core log of the Cerattepe-Artvin area, modified after [31].

3.2.2. Gossan Zone

This gossan material formed as a result of supergene processes associated with uplift and oxidative weathering of the ores. The gossan contains secondary lead, barite, and small amounts of fine-grained free gold and plumbojarosite. This zone can be divided into six parts: Limonitic gossan, gray (barytic) gossan breccia, bedrock gossan, gossan hanging wall-rock sediments, and cover gossan. The boundaries between different types of gossan are usually transitional.

Limonitic gossan constitutes 45% of gossan material. This zone forms a mantle on the sulfide structure and formed by the degradation of the sulfide minerals because of oxidative weathering. The limonitic zone contains rusty brown, fine grain, rich in clay, rock fragments.

Gray gossan has formed on the southern end of the mine deposit and it shows more continuity than others with its thickness varying between 5 and 15 m. It is seen as small lenses intertwined with limonitic gossan in the center of the mineral deposit and in the northern part. Gray gossan has a sandy structure with yellowish brown-light gray colors with low iron content (Figures 8 and 9). Since it contains a significant amount of barite, its specific mass is higher than other gossan types. Its gold and silver content is generally high. It can be said that the gray gossan is formed by the alteration of the barite and sphalerite parts of the mineral deposit. Gossan breccia is usually located in the middle of the ore deposit between limonitic gossan and the main rock. It has a bright rusty red color and contains a dense amount of hematitic and siliceous main rock parts. Siliceous rock fragments are in the goat matrix. Breccia gossan does not show continuity and it is transitive between limonitic gossan and main rock gossan (Figure 8). It has an irregular distribution of gold grade. 20 g/t is observed at high-grade sections.

The bedrock gossan was formed by the alteration of the siliceous, pyritic main rock. There are two sections in this zone. The first section is the part where the main rock has decayed in its place and the second section is talus which extends towards the upper edge of sulfidic and limonitic gossan zones that are adjacent to the main rock. This gossan unit shows continuity throughout the silica main rock adjacent to the sulfide deposit. The average gold grade is usually less than 0.5 g/t. When hanging wall-rock sedimentary rocks and lapilli tuffs are over or inside the limonitic and gray gossan zones, they are generally very rusty. Although not showing continuity, the gossany parts of the hanging wall rock often contain more than 1.0 g/t of gold. The cover formed at areas where gossan has economical gold content. The cover is rich with clay and contains big rock parts belonging to the hanging wall and the base.

| MINERAL | STAGE OF MINERALIZATION | | | | |
|---|---|---|---|---|---|
| | I | II | III | Supergene Enrich. | Oxidation |
| Pyrite | ▬▬▬ | ─ ─ ─ | ─ ─ ─ | | |
| Chalcopyrite | ▬▬ | ─ ─ | ▬▬ | | |
| Marcasite | | ▬▬ | | | |
| Sphalerite | | ▬▬ | ─ ─ ─ | | |
| Galena | | | ▬ | | |
| Tenantite tetrahed. | | | ▬▬ | | |
| Bornite idaite | | | | ▬ | |
| Gold | | | ─ | | |
| Silver | | | ─ | | |
| Covellite digenite | | | | ▬▬ ··· | |
| Chalcosite | | | | ─ | |
| Cuprite | | | | ─ | |
| Cubanite | | | | ─ | |
| Barite | | ───────────── | | | |
| Quartz | | ──── | | | |
| Hem/Um/Jaros.Mal/ Azur./Lepid. | | | | | ── |

**Figure 6.** Cerattepe Cu-Au (±Zn) Mineral paragenesis after [41].

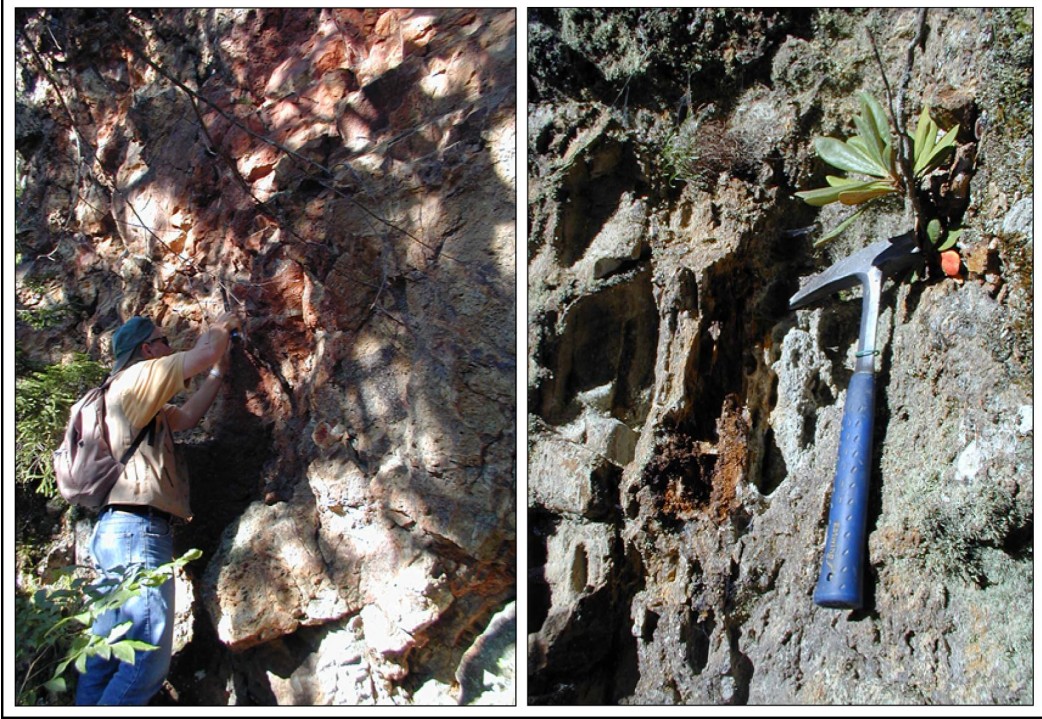

**Figure 7.** Close-up field view of gossan zone at Cerattepe VMSD, Artvin.

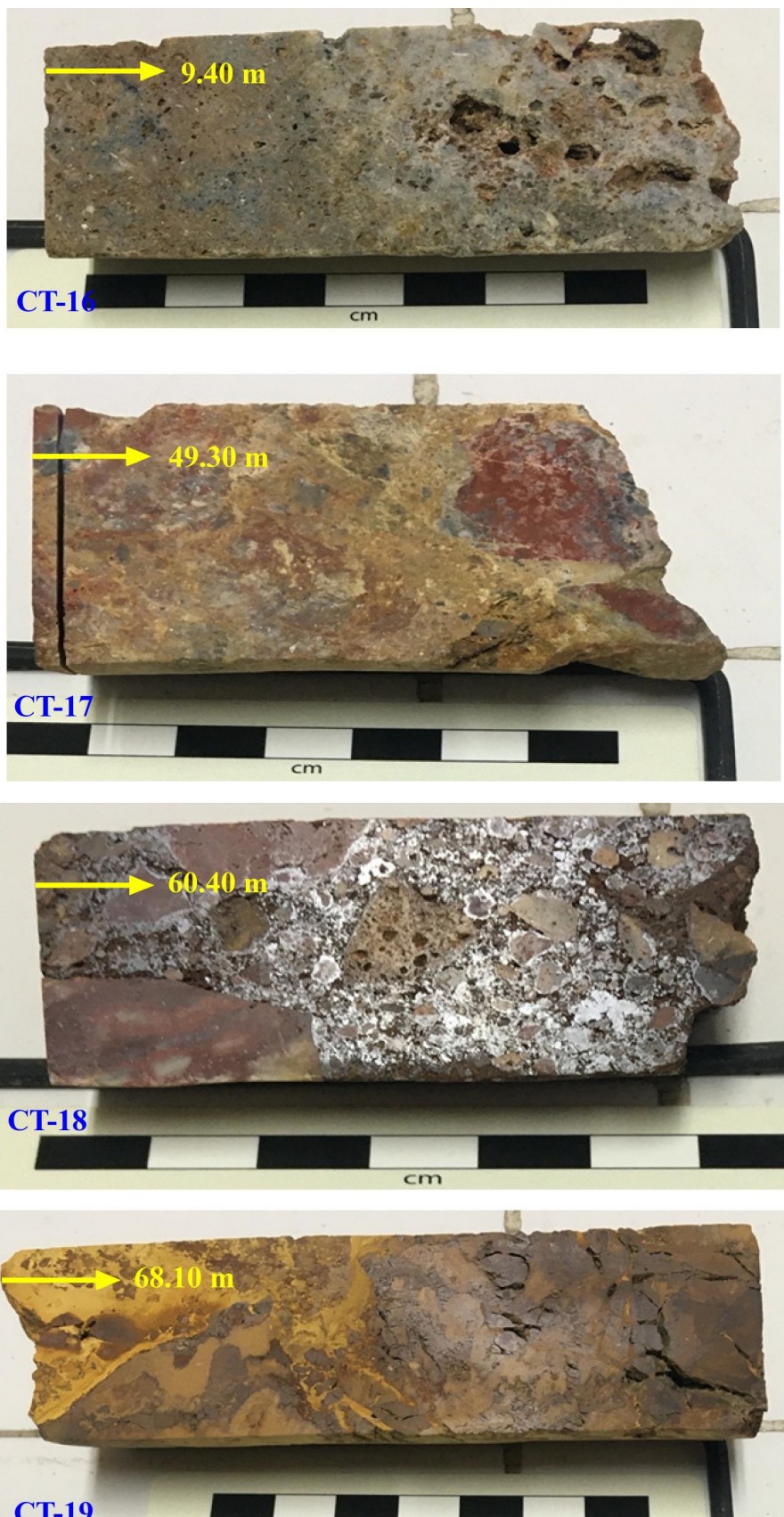

**Figure 8.** Close-up view of polished well log samples from gossan zone of Cerattepe VMSD, Drill core # CTD-190, at 1655 m.

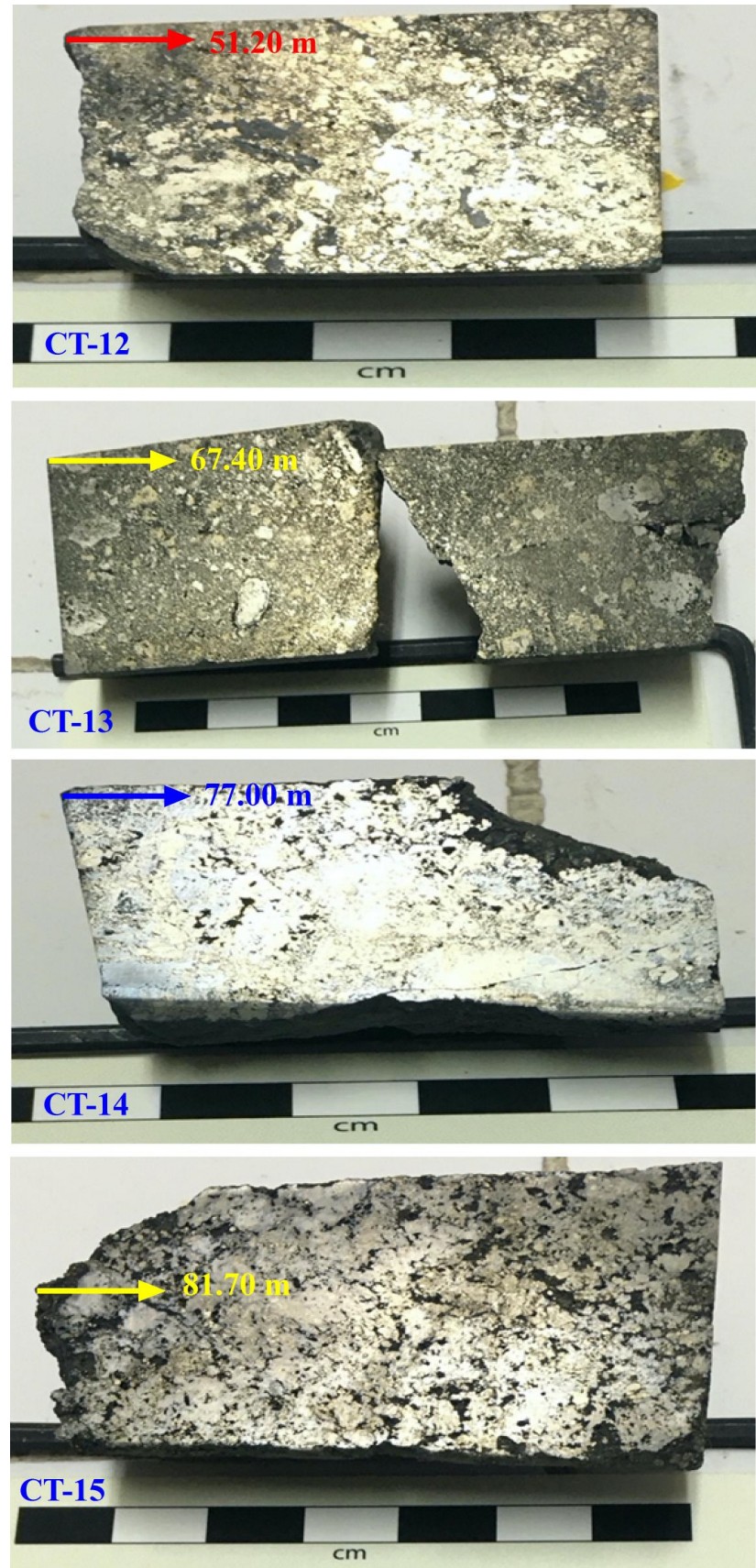

**Figure 9.** Close-up view of polished well log samples from Cerattepe VMSD, Drill core # CTD-117 at 1645 m.

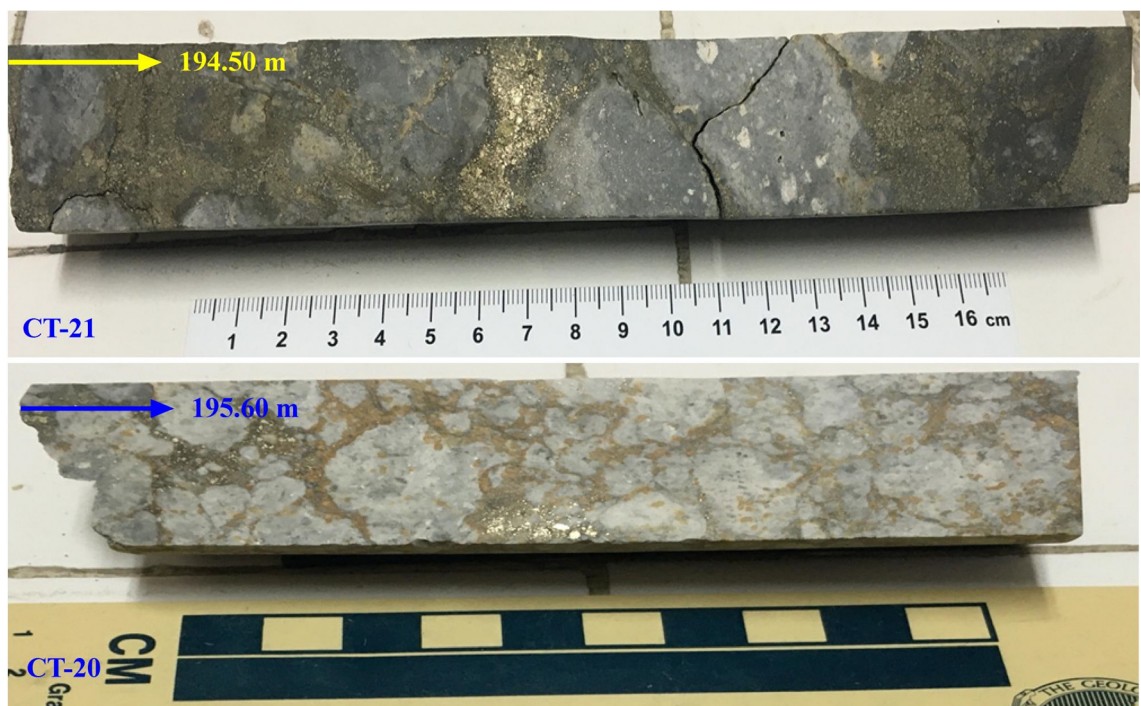

**Figure 10.** Close-up view of polished well log samples from Cerattepe VMSD, Drill core # CTD-111 at 1660 m.

The gossan mineralization zone shows a coarse volcanoclastic structure preserved in a fine crystalline sulfide rock. The common features of this breccia are that they are porous and hollow. Pyrite is intact, but most of the remaining copper is in the form of secondary sulfides. This low-grade zone contains more sphalerite and galena than the other one. A vein zone with a thickness of 1 to 15 m showing continuity has formed inside the felsic tuffs just below the massive sulfides. This zone contains pyrite and a small amount of chalcopyrite and chalcocite-covellite. Intensive kaolinite alteration is observed in tuffs surrounding the veins.

## 4. Results and Discussion

### 4.1. Re-Os Geochronology

The timing of ore deposition was evaluated directly with Re-Os geochronology of the sulfides. Re-Os provides also a tracer isotopic ratio, $^{187}Os/^{188}Os$, which informs of the ore fluid source and/or its history. The Cerattpe (in Pontides) dated with Re-Os age 62 ± 3 Ma (Table 1, Figure 11).

**Table 1.** Re, Os concentrations $^{187}Re/^{188}Os$, 187Os /$^{188}Os$ ratios with error margin of massive sulfide ores from drill Core samples of Cerattepe VMSD.

| Sample | Re (ppb) | Os (ppt) | $^{187}Re/^{188}Os$ | Error | $^{187}Os/^{188}Os$ | Error | Mineral |
|--------|----------|----------|---------------------|-------|---------------------|-------|---------|
| CT-21 | 25.21 | 142 | 1051 | 52. | 1.82 | 0.09 | py ± cp |
| CT-14 | 143.7 | 268 | 5105 | 255 | 594 | 0.30 | py + cp ± cv |
| CT-13 | 94.95 | 152 | 3029 | 151 | 372 | 0.19 | py + cp |
| CT-16 | 9.47 | 121 | 403 | 20 | 1.10 | 0.06 | sulfide rich-silicified whole rock |

The radiogenic Os initial value calculated by the isochron ($^{187}Os/^{188}Os$ = 0.66 ± 0.1) clearly shows a crustal source for Os and by inference, the metals present at the deposit. Since no measurements of Re-Os isotopic composition of the surrounding igneous or sedimentary rocks exist, relating a specific source for the metal is not possible.

Several studies have provided Re-Os ages in Turkey, but are quite limited [42–48] (Figure 12). In the Pontides, two studies show that Cyprus type mineralization in Kure deposit [48] in the northern Pontides occurred approximately 170 Ma, and porphyry copper deposits in the area span a range of 60–90 Ma [45]. VMSD in the eastern Pontides is typically interpreted as Late Cretaceous [13,16,17,19,49], whereas multiple porphyry mineralization events took place from Early Cretaceous to the Early Eocene [45] (Figure 12).

In this area, the geodynamic evolution of the eastern Pontides orogenic belt remains contested in large part because of the lack of systematic geologic, geochemical, geochronological, and geophysical data. The favored theory is that the eastern Pontides orogenic belt was shaped by the northward subduction of oceanic lithosphere (Neotethys Ocean) and the VMS deposits being studied are Kuroko-type deposits that were formed in the back-arc basin environment [4,13,17,18,49,50].

A different view presented by Eyuboglu et al. [16] is that the VMS deposits, located in large-scale caldera-like depressions, where are closely related to the dome-like structures of felsic magmas that are found in these depressions. The main fracture systems of the eastern Pontides orogenic belt controlled their distribution, and it is theorized that the VMS deposits were generated either in an intra arc or near arc region (in the front of arc axis) of the eastern Pontides orogenic belt during the southward subduction of the Tethys oceanic lithosphere.

However, none of the geodynamic models predict or imply specific mineralization ages in EPTB, and studies on the age of the VMSD in EPTB are limited [13,14,16,17,51]. For instance, to date the deposits [19] used lead isotope ratios measured on galena for three major VMS deposits in the region [19], the Murgul main ore body, Lahanos, and Köprübaş.

Eyuboglu et al. [13] posit that the VMS deposits in the northern part of the eastern Pontides orogenic belt occur in two different stratigraphic horizons consisting primarily of felsic volcanic rocks from the Late Cretaceous sequence. The geochemical characteristics of subduction-related calc-alkaline and shoshonitic magmas were revealed by the felsic rocks of first and second horizons, respectively, in continental arcs and represent the immature and mature stages of a Late Cretaceous magmatic arc. As Turonian obtained in his study, SHRIMP zircon U–Pb analyses for the Kizilkaya Formation or ore-bearing dacites hosting the Murgul, Cayeli, Kutlular, Lahanos, Harköy, and İsrail VMS deposits shows a weighted mean $^{206}Pb/^{238}U$ age of 91.1 ± 1.3 Ma. The Çayırbag or Tirebolu Formation hosting the Koprubasi VMS deposit was dated as late Santonian and early Campanian with zircon ages ranging between 82.6 ± 1 Ma and 86.6 ± 0.8 Ma.

One of the deposits in EPTB, VMSD near Tunca, the U-Pb LA-ICP-MS zircon dating of a dacitic tuff breccia yielded an age of 88.1 ± 1.2 Ma (Coniacian-Upper Cretaceous); this was interpreted to be the age of the sulfide occurrences [49].

Our study in the Cerattepe with Re-Os age 62 ± 3 Ma is younger than understood previously, which opens up new areas for exploration. A combination of the Re-Os mineralization here and the alteration silicate ages reported in Murgul [52] and in Kizilkaya [53] implies that mineralization in the area occurs at a distinctly younger interval.

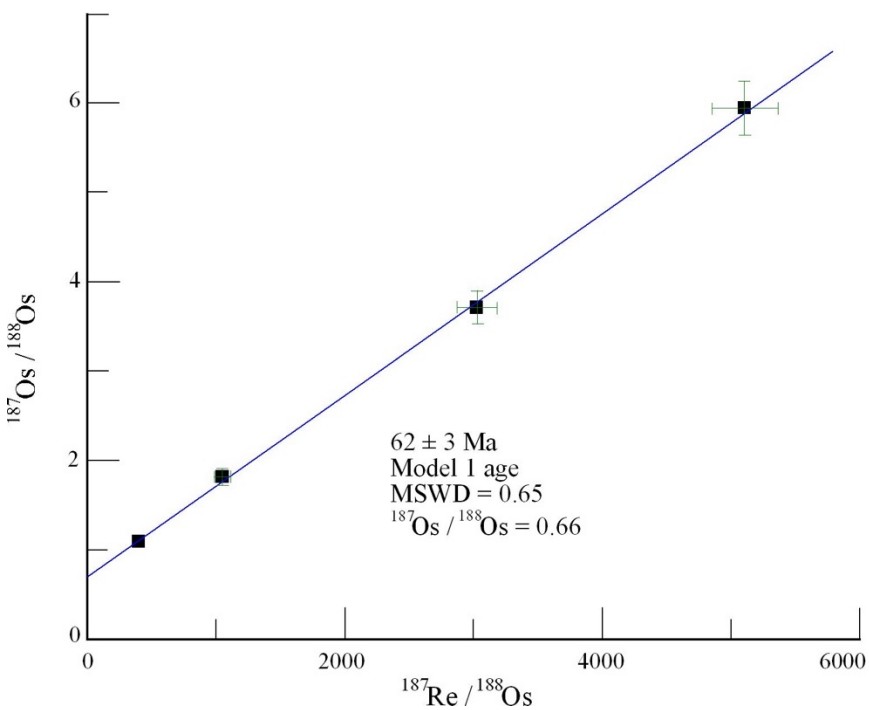

**Figure 11.** $^{187}$Os /$^{188}$Os vs. $^{187}$Re /$^{188}$Os isochron diagram of massive sulfide ores from drill Core samples of Cerattepe VMSD, Re-Os isochron age with 62 My ± 3 Ma.

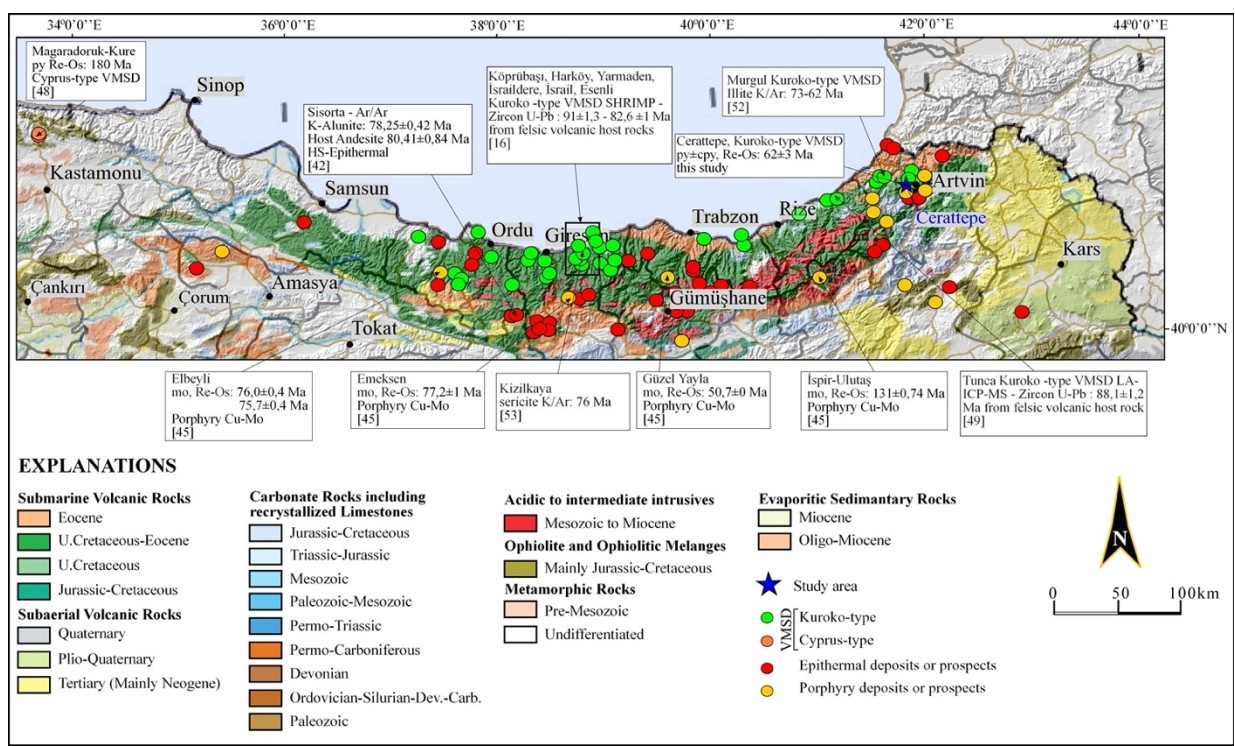

**Figure 12.** Geological map with continuous SRTM topography surface and distribution of the VMS, Epithermal, and porphyry Cu-Mo deposits and prospects of North Eastern Turkey with emphasis on host-rock lithology. U-Pb, Re-Os, K-Ar and Ar/Ar ages of selected deposits, and location of Cerattepe VMS deposit, host-rock lithology is after [17].

### 4.2. Stable Isotopes

VMS deposits are base-metal (Fe, Cu, Zn, Pb, Au, and Ag) sulfide deposits; these are normally stratiform but are rarely stockwork that developed at or just beneath the seafloor in marine settings. The sources of ore fluids (H, C, O, and S), the temperatures of ore deposition environment (O and S), the conditions of silicate alteration (O), the channels of fluid movement (O), the fluid origin (H, O, and S), the redox variations (C, S, and Fe), and fluid phase-separation (O and H) processes have been detected with the use of stable isotopes. To understand the overall isotopic variability in modern as well as ancient VMS deposits, nontraditional stable isotopes (Ca, Fe, Cu, Zn, Se, and Hg) have been used to determine the causes of isotopic fractionations, and the sources of metals [54–58].

#### 4.2.1. O-H Isotope

The isotopic composition of evolving mineralizing fluids in VMS deposits are plotted in the $\delta D$–$\delta^{18}O$ space (Figure 13, Table 2) and cluster near seawater values. Processes such as water-rock interaction, evaporation, boiling, and mixing produce the isotopic pathways revealed on this diagram.

Oxygen and hydrogen isotope analysis of quartz minerals that have been simultaneously formed with mineralization as taken from the oxide zone over the Cerattepe VMS deposit is given in Table 2. The origin of water in the hydrothermal solution that is effective in the formation of oxide zone over the Cerattepe VMS deposit shown in Figure 10 is meteoric-hydrothermal with its values varying between $\delta^{18}O$ = +7.93 to 10.82‰, $\delta D$ = −78 and −68‰.

**Table 2.** $\delta^{18}O$-$\delta D$ isotope data of samples from gossan zone of Cerattepe VMSD.

| Sample No. | $\delta^{18}O$ | $\delta D$ | Mineral |
|:---:|:---:|:---:|:---:|
| CT-1 | 10.48 | −71 | Quartz |
| CT-2 | 9.47 | −74 | Quartz |
| CT-3 | 7.93 | −69 | Quartz |
| CT-4 | 8.42 | −68 | Quartz |
| CT-5 | 10.82 | −73 | Quartz |
| CT-8 | 10.21 | −70 | Quartz |
| CT-9 | 10.01 | −78 | Quartz |

The reported data generally indicates that fluid inclusions from VMS deposits are found in decreasing order within quartz, calcite, barite, anhydrite, sphalerite, enargite, pyrite, and opal. The FI from VMS deposits are mostly two-phase, liquid-plus-vapor inclusions that homogenize to the liquid phase with salinities; most of these are less than 5–10 wt% and low (mostly undetectable) concentrations of $CO_2$ or other gases. Akpinar et al. [42] found two phases of fluid inclusion in quartz, gypsum, calcite, barite, and sphalerite minerals in the Cerattepe VMS deposit including a liquid phase ($H_2Ol_{liquid}$ + $H_2O_{gas}$) with Th = 130–354 °C and salinity NaCl wt% eq. = 0.18–9.47 and a carbonic phase ($H_2O_{liquid}$ + $CO_{2liquid}$ + $CO_{2gas}$) with Th = 238–467 °C and NaCl wt% eq. = 0.21–11.79 [59], compatible as according to [60–63].

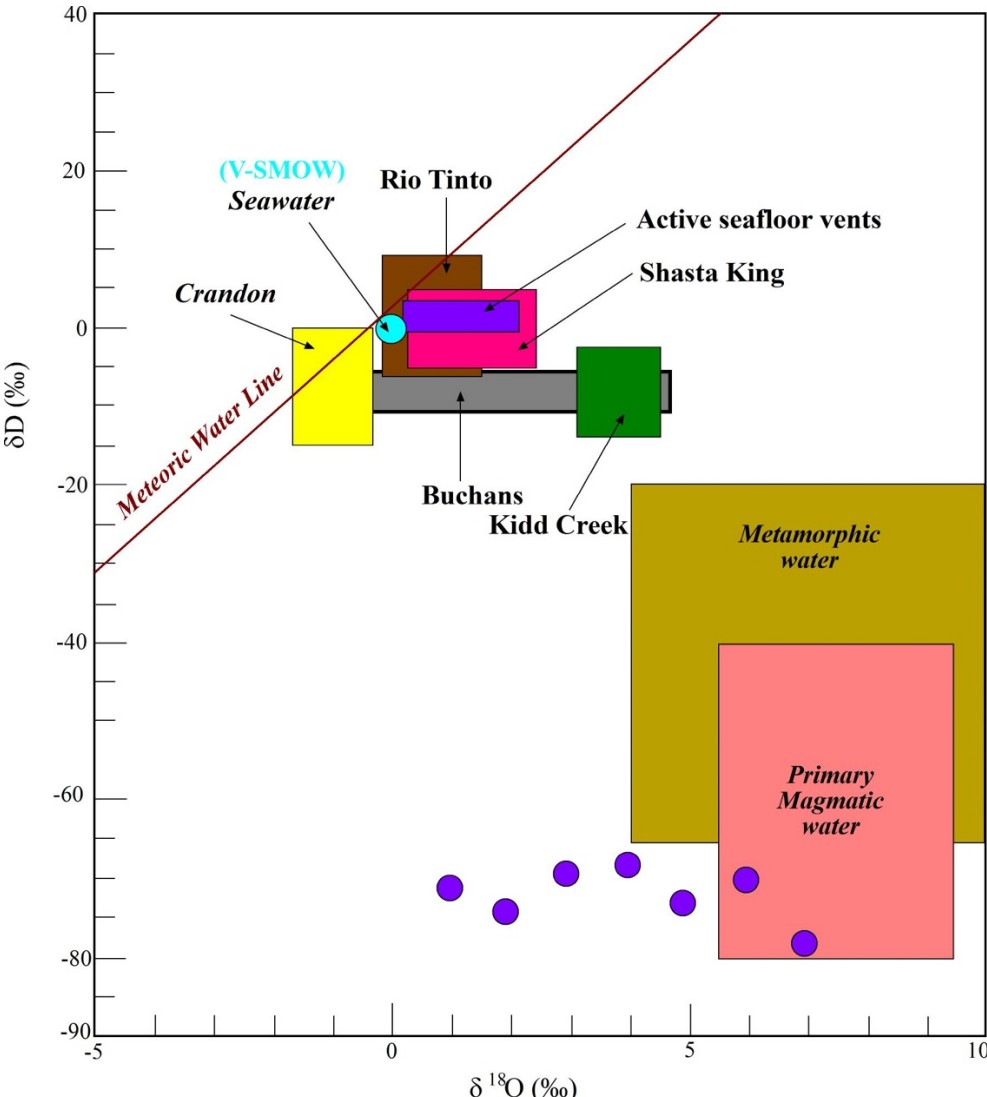

**Figure 13.** $\delta^{18}$O-$\delta$D diagram of samples from gossan zone of Cerattepe VMSD, and comparison with well-known VMSD [40]. With metamorphic water, primary magmatic water area [64], and meteoric water line [65].

4.2.2. S isotope

Numerous studies have focused on sulfur isotopes in VMS deposits according to the summaries [40,66]. There is a fundamental correlation between the sulfur isotope values of stratiform sulfide deposits (including Sedex deposits) and coeval seawater sulfate which was first recognized by Sangster [67]. Huston [66] combined existing data for sulfide and sulfate minerals from VMS deposits with the seawater sulfate $\delta^{34}$S data [68,69], based on evaporite sulfate minerals) over the past 1 b.y. These convective data support the conclusions of Sangster [67] that sulfides in VMS deposits have average $\delta^{34}$S values that are 17.5% less than that of coeval seawater. The data also clearly indicates that individual districts can have a large range of $\delta^{34}$S values: however, the general correlations indicate that a genetic link exists between the seawater sulfate incorporated in convective fluids and sulfide derived by sulfate reduction in the hydrothermal systems. Because of the remobilization of bacterial sulfides or enhanced sulfate reduction involving carbonaceous material in sediments, in general, the range of $\delta^{34}$S values for modern seafloor sulfide deposits is extended to sediment-covered ridges.

Large variations in $\delta^{34}$S suggest that there may be multiple sources for the sulfur or changes in the depositional temperatures. In addition to the large variations in the $\delta^{34}$S

values, the ubiquitous presence of sulfates in Zn-Pb-Cu type VMS deposits suggests that a large amount of the sulfur comes from seawater.

The $\delta^{34}$S values for most igneous rocks are around $0 \pm 5\%$ [70,71]. However, primarily due to changes in the original composition, there are regional variations from this range. The seeming similarity between the $\delta^{34}$S values of igneous rocks and those of the ore deposits indicates that large amounts of the sulfur came from the igneous rocks. However, it is also true that such composition of the $\delta^{34}$S could be created by mixing of the seawater and the pristine ore fluids or the inorganic reduction of seawater sulfate or equilibrium reactions between the circulating seawater and the basement rocks because of a close stratigraphic association of the deposits with the magmatic rocks in the region.

Sulfur isotope values relating to Cerattepe surface and core samples are given in Table 3, Figure 14. While $\delta^{34}$S ratios of all surface rock samples vary between $-3$ and $+12\%$, in the sulfur minerals that are differentiated from the drill core samples, this ratio varies between $-5$ and $0\%$. These values are important since the origin of S points out to magmatic origin. The fact that $\delta^{34}$S exhibits a wide range in the oxide zone ($\delta^{34}$S $= -3$ and $+12\%$), can be explained as differentiation/fractioning of S isotope under the oxidation conditions.

**Table 3.** $\delta^{34}$S isotope data of samples from gossan and massive ore zone of Cerattepe VMSD. Each color represents different drill core and gossan samples.

| Sample No. | $\delta^{34}$S | Nature of Samples, Depth from Surface | Minerals |
|---|---|---|---|
| CT-4 | 11.70 | Outcrop-Gossan | Py |
| CT-6 | 3.60 | Outcrop-Gossan | Py |
| CT-7 | $-3.20$ | Outcrop-Gossan | Py |
| CT-11 | 3.10 | Outcrop-Gossan | Py |
| CT-12 | $-1.6$ | CTD-117, 51.20 m | Py |
| CT-13 | $-2.5$ | CTD-117, 67.40 m | Py |
| CT-14 | $-4.9$ | CTD-117, 77.00 m | py + cp $\pm$ cv |
| CT-15 | 0.4 | CTD-117, 81.70 m | Whole Rock |
| CT-20 | $-0.4$ | CTD-111, 195.60 m | py + cp $\pm$ bn $\pm$ cv |
| CT-21 | $-1.0$ | CTD-111, 194.50 m | py $\pm$ cp |

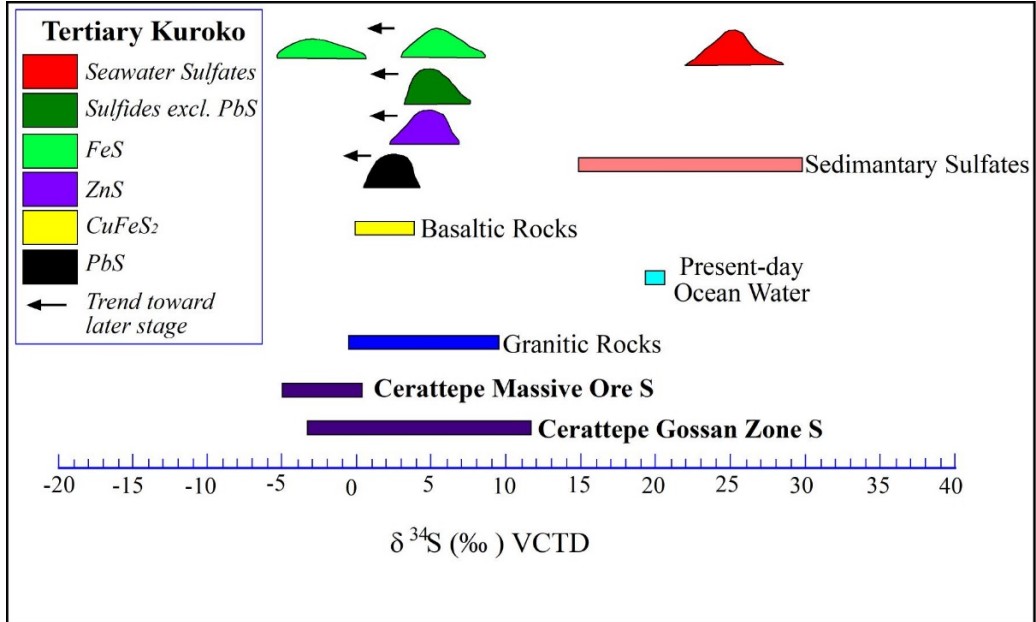

**Figure 14.** $\delta^{34}$S diagram of Cerattepe VMSD with common geologic environments [71] and comparison with Tertiary Kuroko types [70].

Sulfur isotope rates of Cerattepe VMS deposit (surface $\delta^{34}S = -3, +12‰$, drill core $-5, 0‰$) show similarity to the isotope rates determined for Kuroko type of VMS deposits [37,40,70,72–76].

### 4.2.3. Cu Isotope

Copper isotope ($^{65}\delta Cu/^{63}\delta Cu$), bears significant importance in understanding metal cycle and oxidation-reduction processes, as being a new and powerful approach. Cu isotope in the supergene environment varied in the interval of gossan zone $^{65}\delta Cu = -1.1–+1‰$ in the leached zone, in the interval of $^{65}\delta Cu = -1–+4‰$ (mean: +1.2‰) in enrichment zone, and an interval of $^{65}\delta Cu = 0–+1.5‰$ in surface water (mean 1.5‰), and in the interval of $^{65}\delta Cu = -1–+1‰$ (mean: +0.1‰) in deep primarily mineralization and that oxidation of primary mineralization could be explained by the dissolution process [77]. Cu isotope distributions of different geological sources have been summarized [77] and it has been stated that primary magmatic rocks had values that were close to 0.

In a typical porphyry Cu deposit where leach caps have developed in a similar way to the gossan present at Cerattepe, $\delta^{65}Cu$ isotope values are lower near surface (at the oxide zone) and enriched by heavy (+) isotopes at the enrichment (supergene)zone. The primary (hypogene) zone it was close to "0" value [78]. Reworking of copper enrichment zone most likely occurred in the area. As seen in sample CT-15 the value of $-3.59‰$ (Table 4) that occurs near the base of the gossan/top of the enrichment. The most likely explanation for this value would be that the enrichment zone was exposed to oxidative weathering and the heavier Cu isotopes were leached via oxidation (Figure 15).

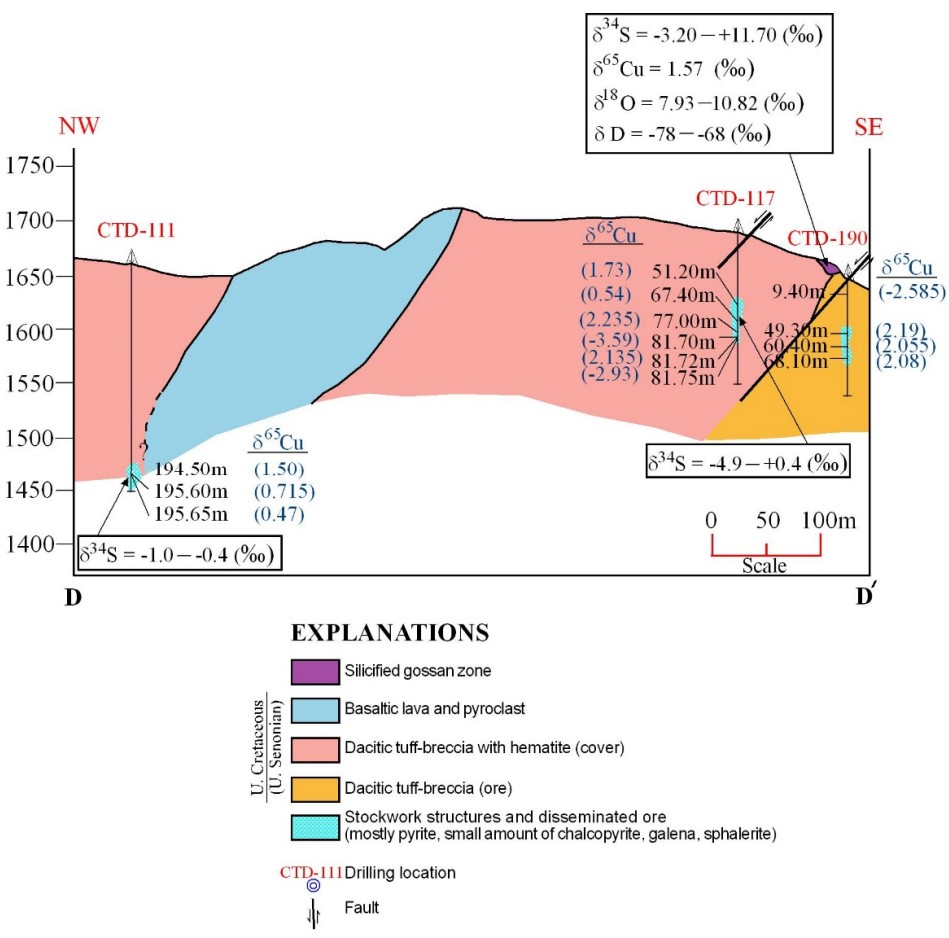

**Figure 15.** D–D′ cross section along drill cores CTD-111, CTD-117, and CTD-190 with reference to Figure 3. $\delta^{18}O$, $\delta D$, $\delta^{34}S$ and $\delta^{65}Cu$, isotope data of samples from drill cores gossan zone of Cerattepe VMSD, Isotope data are from Tables 2–4 respectively.

**Table 4.** $\delta^{65}$Cu isotope data of samples from gossan and massive ore zone of Cerattepe VMSD. Each color represents different drill core samples.

| Sample # | Explanations | $\delta^{65}$Cu 1st Measurement | $\delta^{65}$Cu 2nd Measurement | $\delta^{65}$Cu Average | Nature of Samples, Depth from Surface |
|---|---|---|---|---|---|
| CT-11 | Whole Rock | 1.62 | 1.53 | 1.575 | Outcrop-Gossan |
| CT-12 | py | 1.78 | 1.68 | 1.73 | CTD-117, 51.20 m |
| CT-13 | py | 0.61 | 0.47 | 0.54 | CTD-117, 67.40 m |
| CT-13 | py+ cp | 0.5 | 0.39 | 0.445 | CTD-117, 67.44 m |
| CT-14 | py+ cp $\pm$ cv | 2.34 | 2.13 | 2.235 | CTD-117, 77.00 m |
| CT-15 | py+ cp $\pm$ cv | −3.47 | −3.71 | −3.59 | CTD-117, 81.70 m |
| CT-15 | py + cp | 2.23 | 2.04 | 2.135 | CTD-117, 81.72 m |
| CT-15 | py + cp | −2.85 | −3.01 | −2.93 | CTD-117, 81.75 m |
| CT-16 | Whole Rock | −2.5 | −2.67 | −2.585 | CTD-190, 9.40 m |
| CT-17 | Whole Rock | 2.19 | 2.19 | 2.19 | CTD-190, 49.30 m |
| CT-18 | Whole Rock | 2.06 | 2.05 | 2.055 | CTD-190, 60.40 m |
| CT-19 | Whole Rock | 2.11 | 2.05 | 2.08 | CTD-190, 68.10 m |
| CT-20 | py $\pm$ sp | 0.62 | 0.81 | 0.715 | CTD-111, 195.60 m |
| CT-20 | py + cp $\pm$ bn $\pm$ cv | 0.47 | 0.47 | 0.47 | CTD-111, 195.65 m |
| CT-21 | py $\pm$ cp | 1.48 | 1.52 | 1.5 | CTD-111, 194.50 m |

Few studies have reported copper isotope values for ores in the area. The copper isotope values measured in the hypogene ores presented here overlap with values reported from chalcopyrite and bornite [42,79,80]. The deposit types presented in these contributions ranges from Kruko-type to vein style mineralization. The data presented here expand the range of copper isotope values associated with ore genesis in the area and highlight that supergene processes have remobilized and concentrated metals in the eastern Pontides ore deposits.

## 5. Conclusions

The combined geochemical approach to characterize the ores and quartz veins from the Cerattepe VMS have revealed the following:

- The Re-Os age 62 $\pm$ 3 Ma in the Cerattepe VMSD at eastern Pontides is younger than previously recognized as Late Cretaceous [13,16,17,19,21]. The radiogenic Os initial value calculated by the isochron clearly shows a crustal source for Os and by inference, the metals present at the deposit. Since no measurements Re-Os isotopic measurements of the igneous rocks or sedimentary rocks exist for this area, relating a specific source for the metal is not possible.
- The origin of water in the hydrothermal solution that is effective in the formation of oxide zone over the Cerattepe VMS deposit shows meteoric-hydrothermal with its values varying between $\delta^{18}$O = 7.93–10.82‰, $\delta$D = −78 and −68‰.
- $\delta^{34}$S ratios of all surface rock samples vary between −3 and +12‰, in the sulfur minerals that are differentiated from the drill core samples, this ratio varies between −5 and 0‰. These values are important since the origin of S points out to magmatic origin. The fact that $\delta^{34}$S exhibits a wide range in the oxide zone ($\delta^{34}$S = −3 and +12‰), can be explained as differentiation/fractioning of the S isotope under the oxidation conditions.
- Copper isotope ratios from the ores in the gossan and massive zones indicate that there has been significant migration and concentration of copper during supergene weathering after the deposit formed and was eroded.

**Author Contributions:** Conceptualization, A.U.; methodology, R.M. and G.B.A.; software, N.O.; validation, A.U., R.M. and G.B.A.; formal analysis, R.M., J.K. and J.R.; investigation, A.U., C.S.D., N.O., T.E. and M.E.; resources, M.E., R.M. and G.B.A.; writing–original draft preparation, A.U.; writing–review and editing, R.M.; visualization, M.E. and N.O.; supervision, A.U.; project administration,

A.U.; funding acquisition, A.U. All authors have read and agreed to the published version of the manuscript.

**Funding:** The funding granted to Ali Ucurum by the Turkish State Planning Organization DPT-2003K120280 in 2003, North Atlantic Treaty Organization NATO-EST-CLG-978716 in 2005, and Sivas Cumhuriyet University Scientific Research Fund CUBAP, M-771 in 2019 greatly appreciated.

**Acknowledgments:** Cominco-Turkey is thanked for allowing us to collect samples during field work.

**Conflicts of Interest:** The authors declare no conflict of interest.

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
