# Peer review of "Re-Os Age and Stable Isotope (O-H-S-Cu) Geochemistry of North Eastern Turkey’s Kuroko-Type Volcanogenic Massive Sulfide Deposits: An Example from Cerattepe-Artvin"

_minerals, doi:10.3390/min11020226_

Round 1
Reviewer 1 Report
I recommend a thorough rewrite to improve clarity and separate the discussion from the results. There are numerous repetitions and long sections of ore and deposit description that are not justified by the new data. Some of this could be included if the manuscript were rewritten to focus on the nature of the deposit.

Reviewer 2 Report
The work under revision displays a number of very interesting new data on VMS deposits in Turkey, including Re-Os and stable isotope values. However, I find that these data are not sufficiently interpreted, so that their even usefulness could be reasonably questioned. Just an example: to me, the authors' conclusion that Cu isotopes indicate that there has been significant migration and concentration of copper during supergene weathering is irrelevant: supergene weathering of VMS (and other deposits) generally involves migration and concentration of a number of metals, including copper.
In general, I conclude that discussion and conclusions should be reworked, as they do not make the data justice.
I do NOT question the usefulness of isotopic data as a very important tool in assessing the evolution of VMS deposits; neither the data quality. However, the authors fail in positively showing the data usefulness in their case. Accordingly, I ask the authors for a more detailed interpretation of the data they present. This must be a requisite in order to publish their work.
Finally, I note that English language also needs to be revised. This should be not difficult in this case.
I enclose an annotated file in which a number of comments (about language and other issues) are marked.

Round 2
Reviewer 1 Report
Please revise to use the phrase isochron diagram or Re-Os isochron in Fig. 9 not 'ploting diagram'. Also, the data for this isochron should be tabulated with clear links between samples and the isochron.
Please edit the manuscript carefully to correct grammar and incorrect terms. For example, the phrase Negative TIMS does not make sense. Certainly, negative ion mass spectrometry is appropriate.
Author Response
Figure 9 caption changed as follows;
"187Os /188Os vs. 187Re /188Os isochron diagram of massive sulfide ores from drill Core samples of Cerattepe VMSD, Re-Os isochron age with 62 My ±3 Ma."
the data for this isochron tabulated with clear links between samples and the isochron
The text was checked and corrected grammar.
Negative TIMS changed as "negative thermal ion mass spectrometry"
Reviewer 2 Report
The authors have significantly improved their manuscript. In particular, they have solved the many inconsistent use of English in their firs version. Accordinly, I think that the new version can be approved, maybe with minor modifications that could be done during the edition process.
Author Response
The text was checked and corrected grammar.